# Combined non-psychoactive Cannabis components cannabidiol and β-caryophyllene reduce chronic pain via CB1 interaction in a rat spinal cord injury model

**Anjalika Eeswara, Amanda Pacheco-Spiewak, Stanislava Jergova ⓘ, Jacqueline Sagen***

Miami Project to Cure Paralysis, University of Miami Miller School of Medicine, Miami, FL, United States of America

* sjergova@miami.edu

**Data Availability Statement:** All behavioral raw data, statistical evaluation data and supplementary figures are available from the Dryad database, link

## Abstract

The most frequently reported use of medical marijuana is for pain relief. However, its psychoactive component Δ9-tetrahydrocannabinol (THC) causes significant side effects. Cannabidiol (CBD) and β-caryophyllene (BCP), two other cannabis constituents, possess more benign side effect profiles and are also reported to reduce neuropathic and inflammatory pain. We evaluated the analgesic potential of CBD and BCP individually and in combination in a rat spinal cord injury (SCI) clip compression chronic pain model. Individually, both phytocannabinoids produced dose-dependent reduction in tactile and cold hypersensitivity in male and female rats with SCI. When co-administered at fixed ratios based on individual A50s, CBD and BCP produced enhanced dose-dependent reduction in allodynic responses with synergistic effects observed for cold hypersensitivity in both sexes and additive effects for tactile hypersensitivity in males. Antinociceptive effects of both individual and combined treatment were generally less robust in females than males. CBD:BCP co-administration also partially reduced morphine-seeking behavior in a conditioned place preference (CPP) test. Minimal cannabinoidergic side effects were observed with high doses of the combination. The antinociceptive effects of the CBD:BCP co-administration were not altered by either CB2 or μ-opioid receptor antagonist pretreatment but, were nearly completely blocked by CB1 antagonist AM251. Since neither CBD or BCP are thought to mediate antinociception via CB1 activity, these findings suggest a novel CB1 interactive mechanism between these two phytocannabinoids in the SCI pain state. Together, these findings suggest that CBD:BCP co-administration may provide a safe and effective treatment option for the management of chronic SCI pain.

## Introduction

Neuropathic pain often becomes a chronic debilitating condition that results from spinal cord injury (SCI), significantly reducing a patient's quality of life [1–3]. To date, standard therapies for SCI pain such as opioids have low efficacy and are encumbered with undesirable side

https://datadryad.org/stash/share/7iAXzVYToOwNVm2XoykKwzgGJ0wQcn1kcBitvwGwuKY; doi:10.5061/dryad.0rxwdbs4p.

**Funding:** This work was supported by the 2019 Research Grants Program of the Consortium for Medical Marijuana Clinical Outcomes Research which is funded through State of Florida appropriations (JS). The funders had no role in study design, data collection and analysis, decision to publish, or preparation of the manuscript.

**Competing interests:** The authors have declared that no competing interests exist.

effects including tolerance formation and addiction [4–7]. Therefore, many novel approaches are currently being investigated to mitigate chronic neuropathic pain. Recently, there has been increased interest in constituents of the *Cannabis sativa* plant as an alternative treatment.

One of the barriers for widespread therapeutic use of cannabis are the effects caused by the major psychoactive component, delta-9-tetrahydrocannabinol (THC), which has led to differing views on its clinical efficacy and safety [8–10]. However, *Cannabis sativa* contains a multitude of other phytocannabinoids such as cannabidiol (CBD) and β-caryophyllene (BCP) which have been shown to have analgesic potential in reducing chronic pain and lack the psychotropic effects associated with THC [11–16]. Further, the use of CBD and BCP are not hindered by the same stringent federal regulations as THC and can both be purchased over the counter (OTC) thereby increasing accessibility to the general public. CBD oils derived from hemp, with undetectable THC, are now widely available. BCP is a sesquiterpene and major component (up to 35%) in the essential oils of *Cannabis sativa*, found in many other plant species, and available for use as a flavor-enhancing food additive. Taken together, this highlights the importance of exploring other phytocannabinoids that have safer therapeutic profiles and are easier to procure.

Numerous preclinical animal studies have assessed CBD and BCP's analgesic potential in a variety of pain models. These studies have demonstrated that both CBD and BCP significantly reduce hypersensitivity in chronic pain states without any overt side effects [17–19]. Past research has demonstrated the therapeutic potential of individually administered CBD and BCP in peripheral neuropathic pain models such as chronic constriction injury (CCI), spared nerve injury (SNI), and diabetic neuropathy [11,14,20–23]. For example, administration of either CBD or BCP was effective in attenuating or reversing the development of mechanical hypersensitivity in the CCI or SNI model, respectively [14,20]. Since CBD and BCP are thought to act via distinct mechanisms, their combination could provide additive or enhanced antinociception. BCP has been hypothesized to act as a CB2-receptor-selective agonist [24–26] to produce pain-reducing effects in several rodent models, including chronic inflammatory and peripheral neuropathic pain [12,26,27]. Further, BCP has been shown to reduce inflammatory cytokines including tumor necrosis factor-α, interleukin-1β, and interleukin-6 [12,28,29]. In contrast, CBD shows little binding to CB1 and CB2 receptors; thus its mechanism is not well understood, but may involve activation of transient receptor potential channels of both vanilloid type 1 (TRPV1) or ankyrin type 1 (TRPA1) [14,30–32], indirect action via inhibition of endocannabinoid degradation [33,34], or serotonergic system activation via 5-HT1A receptors [14,35,36]. The goal of this study was to assess the potential use of these cannabis components to reduce SCI neuropathic pain. In addition, while tested individually, no studies have analyzed the pain-relieving efficacy of these two phytocannabinoids in combination. To address this, we first evaluated the individual analgesic efficacy of CBD and BCP in a rat chronic SCI pain model. We then explored whether the analgesic potential could be improved through co-administration of these cannabis constituents. Using an isobolographic approach we sought to determine whether additive/synergistic interactions between CBD and BCP result, and whether this combination strategy can provide increased attenuation of SCI-related neuropathic pain. Preliminary findings from this work have been reported previously [37].

## Methods

### Animals

Male and female Sprague-Dawley rats (approx. 140–200 g, Envigo, MN) were used for the experiments. Animals were housed two per cage with corncob bedding and allowed free access to food and water in a 12-h light/dark cycle. Experimental procedures were reviewed and

approved by the University of Miami Animal Care and Use Committee and followed the recommendations of the 'Guide for the Care and Use of Laboratory Animals' (National Research Council).

## Spinal cord injury

The method to induce SCI via clip compression [38] has been used successfully by our laboratory for pharmacologic antinociceptive evaluations and hypersensitivity evaluations over the past several years [39–46]. For all surgeries, aseptic surgical techniques were used. Rats were anesthetized with 4–5% isoflurane in $O_2$ and maintained on 2–3% isoflurane/$O_2$. The back of the rats, from lumbar to cervical vertebrae, were shaved and the skin was swabbed with antiseptic solution. Following incision of the skin, 2–3 thoracic vertebrae were removed and a laminectomy was performed to expose spinal cord segments T6-T8. An aneurysm clip 1 mm wide (20 g compression force; Harvard Apparatus) was oriented in a vertical position on an exposed spinal cord segment between T6-T7. The dura and spinal nerve roots were not disturbed and the clip was left in place for 60 s. The clip was removed, and the surgical area closed. Rats recovered in their home cages and were given free access to food and water. Following spinal compression, bladders were expressed a minimum of three times daily for 7–10 days or until voiding was regained. All behavioral testing began 4 weeks post-SCI once pain behaviors were fully expressed.

## Drugs administration

In order to utilize a readily available OTC source of CBD, Broad Spectrum CBD Gold Oil (Koodegras, Millcreek, UT) was used. BCP was obtained from Sigma-Aldrich (St. Louis, MO). The CB1 and CB2 receptor antagonists AM251 and AM630, respectively, the mu-opioid receptor antagonist naloxone, and mixed CB1/CB2 synthetic agonist WIN 55,212–2 were obtained from Sigma-Aldrich (St. Louis, MO). On each day of the experiment CBD was prepared in a 3:1:16 ethanol/Tween 80/0.9% NaCl plus 2% Tween vehicle and BCP was prepared in a 5% Tween in saline vehicle. CBD or vehicle were administered as an i.p. injection (volume = 0.3ml). BCP or vehicle was administered by oral gavage (18 oral feeding needle, volume = 0.3ml). For antagonist experiments, AM251, AM630, naloxone or vehicle were administered as a s.c. injection 0.5h prior to cannabinoid delivery. In the side effects assessment, positive control WIN 55,212–2 was administered s.c. 0.5h prior to testing. Morphine sulfate (Sigma) was prepared in saline and administered s.c. 30 min prior to CPP training. Additionally, although the focus of the study was assessment of OTC sources, since CBD oils contain trace amounts of minor cannabinoids, a small pilot comparison was done with known CBD (NIDA Drug Supply Program).

## Experimental design

All behavioral measurements were taken prior to SCI and immediately before drug administration at 4 weeks post SCI when animals demonstrated stable pain-related behavior, and then over a 3 week period with at least 72 hr wash out period between dosing as described in individual experiments below. The number of animals per group was determined at the beginning of the study by SigmaStat Power Analysis with the input data based on our previous studies and desired power set at 0.8 with alpha 0.050. For all experiments, animals were randomly assigned to the experimental groups and the experimenter was blind to all drugs or dose combinations being tested.

*Experiment 1—time course and dose response profiles.* To determine time course and dose response profiles, CBD and BCP were individually administered at various doses.

Animals were randomly assigned a dose of either CBD or BCP (or vehicle), tested, then allowed a 72 hr washout period before another dose administration and testing, until sufficient data was attained for each drug/dose. Tactile paw withdrawal threshold (PWT) measurements and acetone cold responses were taken at 0.5, 1, 1.5, 2, and 5 hr post CBD administration. PWT and cold responses were tested similarly but only up to 2 hr post BCP since all significant antinociceptive effects of BCP were resolved by that time point. A50 values, a concentration of a drug needed for half-maximal effect, for individual drugs were calculated (JFlash).

*Experiment 2—fixed-ratio combinations.* To determine the analgesic potential of CBD/ BCP coadministration compared to individual administration and to determine the optimal combination, various fixed-ratio combination doses. based on Experiment 1, were tested. Animals were randomly assigned a dose of CBD/BCP (or vehicles), tested, and then allowed a 72 hr washout period before another dose administration and testing, until sufficient data was attained for each dose combination. Since the A50 doses for both CBD and BCP, obtained from Experiment 1, differed for cold and tactile hypersensitivity, the cold and tactile combination evaluations were performed on different days in the combination studies, with a 72 hr washout period between each administration. Tactile PWT measurements and acetone responses were taken at 0.5, 1, 1.5, 2, and 5 hr post combination drug administration. An additional measure for tactile PWT was taken at 24 hr post administration in case of some residual antinociceptive effects at 5 hr.

*Experiment 3—side effects.* To determine if CBD and BCP produce adverse cannabinoidergic effects, CBD and BCP were in combination at the highest therapeutic dose combination to maximize detection of any potential adverse side effects. Animals were tested with either CBD/BCP or synthetic CB agonist WIN 55,212–2 as a positive control, with 72 hr washout between treatments. To fully evaluate these cannabinoidergic side effects, intact non-SCI animals were needed due to physical limitations following SCI. However, a subgroup of SCI animals was also used to test some side effects when feasible. Rotarod latency was measured at 0.5, 1, 1.5, 2 and 24 hrs. Body temperature and catalepsy were measured at 0.5, 2, 5, and 24 hrs.

*Experiment 4 –antagonists.* To determine potential antinociceptive mechanisms of the CBD/BCP combination, antagonists were administered 30 min prior to CBD/BCP coadministration at their combination A50 dose determined for each test from Experiment 2. Animals were randomly assigned an antagonist, with a 72 hr washout between antagonists until sufficient data was attained. Tactile PWT and acetone response measurements were taken at 0.5, 1, 1.5 2, 5 and 24 hr post drug administration.

*Experiment 5—conditioned place preference.* To determine if repeated CBD/BCP administration can potentially decrease opioid seeking behavior, a conditioned place preference (CPP) test was used with 2x combined A50 dose. This dose was used in order to assure maximum antinociceptive benefits were achieved. Morphine or saline was administered 30 minutes prior to placing the animal in a place preference box and CBD/BCP was administered 1.5 hours prior to coincide with the time of peak antinociceptive effects.

## Behavioral analysis

Two standard sensory tests for tactile hypersensitivity using von Frey filaments and cold hypersensitivity with hindpaw acetone droplets were used. Behavioral testing for tactile and cold hypersensitivity was carried out by individuals blinded to the experimental groups. All testing was done during the light cycle (6am-6pm) in a designated animal testing room.

**Tactile hypersensitivity.** For assessment of mechanical hypersensitivity, calibrated von Frey filaments ranging from 0.4 to 15 g were used [47]. Animals were placed in a clear plastic cage on an elevated wire mesh surface and allowed 15 min to acclimate. The Dixon up-down

method was used and filaments were applied to the right plantar hind paw and kept in place for 6 s [48]. If a response was evoked the next lower filament was tested; if there was a negative response the next higher filament was tested. This process was repeated until a total of six responses were recorded. A positive response was recorded as a brisk withdrawal in conjunction with a supraspinal response to reduce potential confounding SCI spinal hyperreflexia. This requires the animal to also vocalize, orient their head towards the stimulus or groom the tested paw. Generally, uninjured rats do not respond to a force of <15 g. As such, an upper limit of 15 g was used in experiments since a greater force may lift the hind paw itself.

**Cold hypersensitivity.** For assessment of sensitivity to a non-noxious cooling stimulus, responses to acetone droplets on the hind paw were measured [49]. A blunted 22 g needle was used to apply 100 µl of acetone onto the lateral margin of the hind paw. Acetone was applied for a total of 5 times with 2 min between applications. Response frequency (%) was calculated by the number of positive responses out of the five trials. In uninjured rats, acetone does not evoke a withdrawal response. Responses were marked positive only when a supraspinal response was observed in addition to paw withdrawal, such as a head turning towards the stimulus, paw licking, or shaking.

**Conditioned Place Preference (CPP).** For assessment of ongoing pain, a subgroup of male rats with SCI and uninjured animals underwent CPP. The place preference apparatus is a two-chambered box with distinct walls. One chamber has black walls and the other chamber has black and white striped walls. Access to either chamber can be blocked by a removable divider. Animals were acclimated to the open two-chambered box for 30min/day for 2 days before training and testing. Morphine (0.3ml, 2 mg/kg, s.c.) was used as a reinforcing analgesic agent, in conjunction with or without the 2xA50 CBD/BCP dose (2.0 mg/kg i.p. and 16 mg/kg oral gavage, respectively) in order to evaluate potential effects of concomitant CBD/BCP administration on morphine-seeking behavior. This low dose of morphine was chosen because it has been shown in other models to produce a CPP in animals with chronic pain but not intact animals [50,51]. Saline (0.3ml, i.p.) was used as a CPP control treatment. Prior to CPP conditioning, SCI animals were divided into one of three treatment groups: morphine, morphine and CBD/BCP or saline. Uninjured animals were used to assess the extent to which this low dose morphine produced CPP in intact controls. For all groups, on day 0, the animal's preferred side was determined by recording the time spent in each chamber. On conditioning days 1–5, morphine was paired with the rat's non-preferred chamber in the morning and saline with their preferred chamber in the afternoon with the divider closed. On day 6, CPP was evaluated by recording time spent in each chamber with the divider removed and CPP scores calculated by subtracting the time spent in the non-preferred chamber prior to training from time spent in that chamber following morphine pairing.

## Side effects

To assess potential side effects common to cannabis, the highest antinociceptive dose combination CBD/BCP was evaluated for common side effects of the cannabinoid "tetrad" test, including locomotor dysfunction, catalepsy, and hypothermia, in comparison with saline controls.

*Body temperature*. Body temperature was measured by infrared touchless thermometer positioned to the left side of the trunk. *Rotarod test*: The effect of cannabis constituents on motor function was assessed with an accelerating rotarod apparatus (Harvard Apparatus) [52]. One day prior to testing, rats were briefly trained on the apparatus to get accustomed to it. The rotarod was accelerated 5–25 rpm over 60 s. Rats that did not fall off the rotarod prior to the 60s cut-off were assigned a latency of 60s. At each time point evaluated, the latency to fall (s)

off the rotarod was recorded. *Catalepsy test*: Using the bar test [53], the forepaws of the rat were placed on a metal bar located 10 cm above a Plexiglass surface. At each time point evaluated, the total amount of time spent immobile was determined by a stopwatch.

**HEK293 CB1 cells.** The HEK293 cell line expressing N-terminus hemagglutinin tagged-CB1 receptor was a gift from Prof. Ken Mackie, Indiana University (Bloomington, IN) for initial studies, and purchased from Kerafast, North Carolina for more recent analyses [54,55]. Cells were plated and maintained in Dulbecco's Modified Eagle Medium: Nutrient Mixture F-12 (Gibco, Thermo Fisher, USA) supplemented with 10% fetal bovine serum (Sigma-Aldrich, MO, USA) and antibiotics (1% penicillin, 1% kanamycin, 0.5% gentamycin; Gibco). Cells were kept in an incubator at 37°C and 5% $CO_2$. After reaching near to 100% confluence, cells were split; media was removed and cells rinsed in Hank's Balanced Salt Solution (Sigma-Aldrich). Cells were then treated with 0.25% trypsin for 3 minutes for complete removal of cells from the flask and then neutralized with media. All cells were transferred to a 15ml tube and spun at 3500 rpm for 5 min at 4°C to pellet the cells. Afterwards, supernatant was discarded, and cells were resuspended in media. Cells were counted using 0.4% trypan blue and transferred either to a 12well plate (50.000 cells per well), or 25ml flask (5–10 million cells) depending on the treatment.

## CB1 redistribution assay

Cells were plated in 12-well plates at 50,000 cells/well and cultured overnight. Wells were treated with vehicle (negative control), 3μM solution of WIN 55,212–2 (Sigma-Aldrich), CBD (0.5mg/ml), BCP (4mg/ml), CBD/BCP (0.5mg/ml:4mg/ml) and AM251 (1mg/ml, Sigma Aldrich) for 20 minutes. Cultures were rinsed with Hyclone phosphate buffered solution (PBS; Sigma-Aldrich), fixed with 4% paraformaldehyde overnight and washed with Hyclone PBS. Fixed cells were immunostained overnight with Alexa 594 conjugated anti-hemagglutinin 1:500 (2μL/mL, Invitrogen, ThermoFisher, USA) to visualize the CB1 receptor [54,56,57]. Excess antibody was removed and cells were washed with PBS and counter-stained with 4', 6-diamidino-2-phenylindole (DAPI) to reveal the nuclei.

## Statistical analysis

All statistical analyses were done in Graph Pad Prism 8.4.2. Data are expressed as mean±SEM with statistical significance taken at $p < 0.05$. Dose–effect curves were obtained by converting the withdrawal thresholds to a percent maximum possible effect (MPE): ((drug threshold-baseline threshold)/(pre-injury–baseline))x100. A50 values and isobolographic analysis were determined using JFlashCalc software (University of Arizona). All behavioral tests were compared using two-way RM ANOVA with Tukey post-hoc test. Conditioned place preference data were analyzed by one-sample t-test with zero hypothesis and Wilcoxon test and one-way ANOVA for group comparisons.

## Results

We first evaluated the time course and dose responses of CBD and BCP individually to determine A50s. At least 3 doses of each agent were tested to generate time course and dose-response curves and calculate antinociceptive A50. Behavioral testing began at 4 weeks post SCI and showed mechanical and cold hypersensitivity responses differed over time after administration of CBD or BCP.

### Experiment 1: Dose-response effects of CBD reduces tactile and cold hypersensitivity

**Tactile hypersensitivity.** CBD was tested at 0.1–5.0 mg/kg i.p., dose range based on previous reports in other rodent pain models [14,23,58]. CBD reduced mechanical hypersensitivity in males in a dose- and time-dependent manner compared to vehicle (overall $F_{(4, 25)} = 24.21$, $p < 0.0001$) with maximal reduction at 60 min post injection for 3 mg/kg ($p < 0.0001$) and 5 mg/kg doses ($p < 0.0001$) compared to vehicle (Fig 1A). Dose 5 mg/kg was significantly more potent compared to other doses at 60–120 mins post injection, with at least $p < 0.05$. Dose 3 mg/kg was significantly more potent than other lower doses at 60 min post injection, with at least $p < 0.01$. The CBD effects tended to return towards pre-drug baseline by 2–5 hours after administration. However, the highest dose 5 mg/kg showed more prolonged effects in reducing mechanical hypersensitivity, still apparent at 2–5 hrs post-injection with at least $p < 0.05$ compared with vehicle. CBD produced significant reduction in mechanical hypersensitivity compared to vehicle in females starting at 30 min post injection for the 5 mg/kg dose ($p = 0.0034$) and peaking at 60 min for the 5 mg/kg ($p < 0.0001$) and 3 mg/kg ($p = 0.0363$) doses. Dose 5 mg/kg was also significantly more potent at 60 min post injection compared to other doses, with at least $p < 0.05$. The CBD effects appeared to be shorter lasting in females, returning towards pre-drug baseline by at approximately 90–120 min after administration. Overall $F_{(4,20)} = 2.404$, $p = 0.0449$ (Fig 1B).

**Cold hypersensitivity.** In males, 0.11 mg/kg dose showed no difference from vehicle throughout the test period. Doses 3 mg/kg and 5 mg/kg were significantly more potent compared to vehicle at 30–120 mins post injection, with at least $p < 0.05$. Dose 1 mg/kg produced significant effects at 120 min post injection compared to vehicle ($p = 0.0447$). Overall $F_{(4,20)} =$

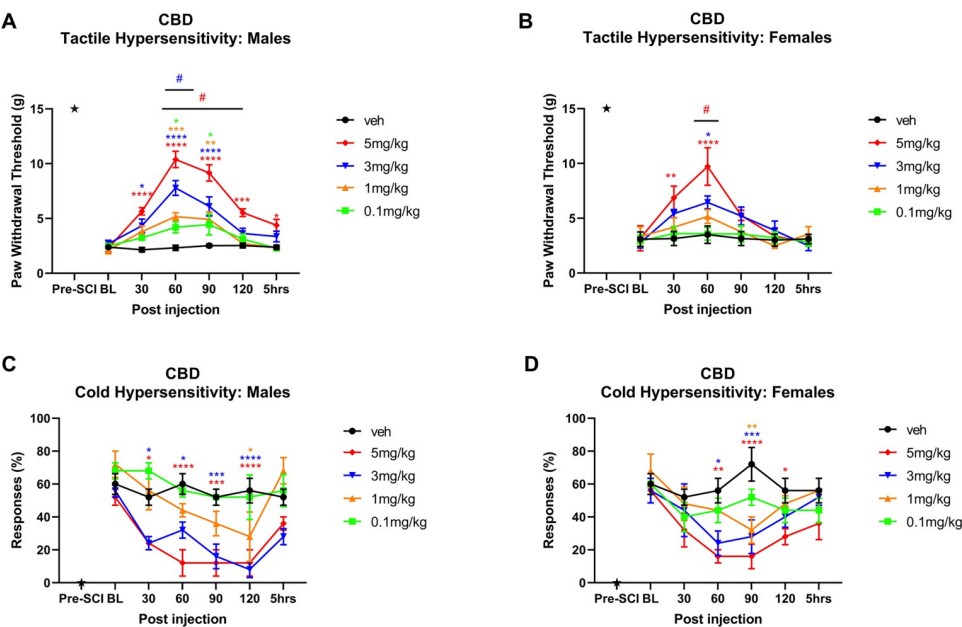

**Fig 1. Time course of CBD antinociceptive effects in male and female rats.** Time plots of the effects of CBD (0.1, 1, 3 or 5 mg/kg) and matched vehicle on mechanical PWT in A) males and B) females, and acetone responses in C) males and D) females (n = 5–6 per treatment group). Animals were tested starting at 4 weeks post-SCI; pre-SCI baseline responses are also displayed as indicated by the star. *, **, ***, **** denote $p < 0.05, 0.01, 0.001$ and $0.0001$ compared to vehicle for each treatment group. # denote p at least $<0.05$ between 5mg/kg (red #) and lower concentrations of CBD or 3mg/kg (blue #) and lower concentrations of CBD.

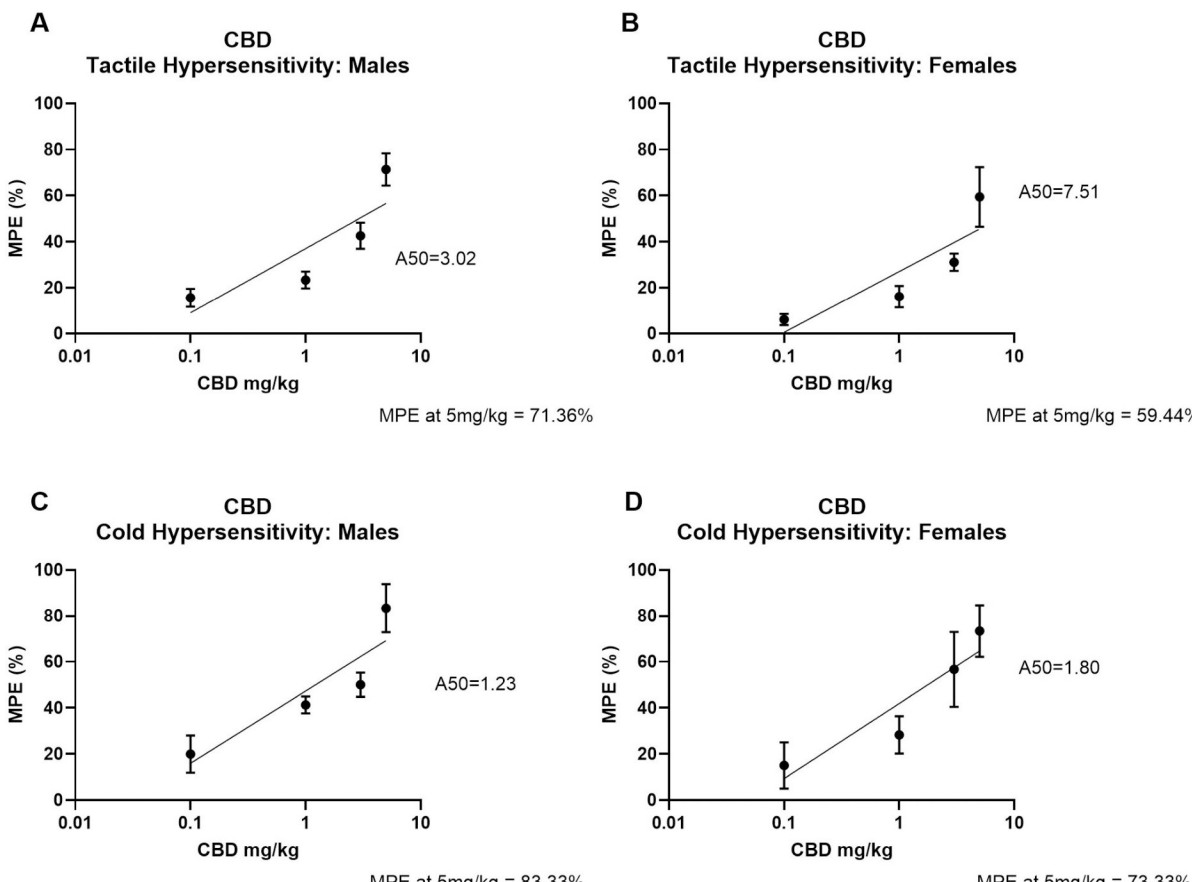

**Fig 2. Dose response curves for CBD antinociceptive effects in male and female rats.** Dose response curves for the effect of CBD on mechanical paw withdrawal threshold in A) males and B) females and acetone responses in C) males and D) females. A50 for each treatment group is displayed. MPE for the highest dose of CBD is indicated as well. Data are shown as % maximal possible effect (% MPE) ± SEM.

14.41, p<0.0001 (Fig 1C). Reduction in cold hypersensitivity was seen in females for the 5mg/kg dose from 60–120 mins post injection (at least p<0.05). Dose 3mg/kg was effective at 60–90 mins (at least p<0.05), dose 1 mg/kg was effective only at 90 min post injection (p = 0015). Overall F(4,20) = 6.043, p = 0.0023 (Fig 1D).

**Dose response.** The MPE for each dose was calculated and used to create dose response plots and to determine A50 for each test (Fig 2). For tactile hypersensitivity, CBD A50 was 3.02 mg/kg in males (Fig 2A) and 7.5 mg/kg in females (Fig 2B). The tactile MPE calculated at 60 mins (approximate time of peak effect) for 5 mg/kg CBD (highest dose used) was 71.36% (males) and 59.44% (females). For cold hypersensitivity, CBD A50s were 1.23 mg/kg (males, Fig 2C) and 1.8 mg/kg (females, Fig 2D), respectively. The cold MPE at 60 mins for 5 mg/kg CBD was 83.33% (males) and 73.33% (females).

Experiment 1: Dose-response effect of BCP reduces tactile and cold hypersensitivity

BCP was administered via feeding tube (oral gavage) at initial dose ranges of 10–50 mg/kg p.o. [27,59].

**Tactile hypersensitivity.** BCP administration in males (Fig 3A) lead to maximum attenuation of tactile hypersensitivity at 60 min post injection at the 20 mg/kg and 50 mg/kg doses (p<0.0001) compared to vehicle (overall F(3,20) = 13.08, p<0.0001). Both doses were also more effective than the 10 mg/kg dose (p<0.0001). Dose 50 mg/kg was also effective at 30 min

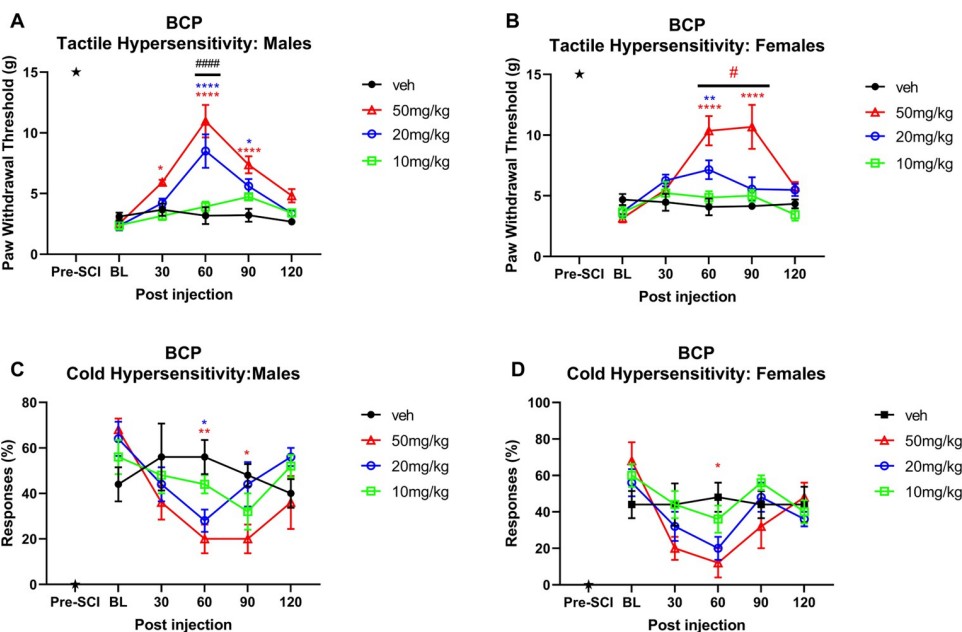

**Fig 3. Time course of BCP antinociceptive in male and female rats.** Time plots of the effect of BCP (10, 20, 50 mg/kg) and matched vehicle on mechanical PWT in A) males and B) females and acetone responses in C) males and D) females (n = 5–6 per treatment group). Animals received a single oral administration at time 0h, 4 weeks post SCI surgery; pre-SCI data is also displayed (indicated by star). *, **, **** denote p < 0.05, 0.01, and 0.0001 compared to vehicle for each treatment group. A) ####p<0.001 for 50mg/kg vs 10mg/kg and 20mg/kg vs 10mg/kg B) #p at least <0.05 for 50mg/kg vs lower BCP doses.

(p = 0.0247) and 90 min post injection (p<0.0001), dose 20 mg/kg was effective at 90 min (p = 0.0112). In females (Fig 3B) a dose of 50 mg/kg showed maximum effect at 60–90 mins post injection (p<0.0001) compared to vehicle. The 20 mg/kg also produced effect at 60 min post injection (p = 0.0051). The 50 mg/kg dose was more potent than all lower doses at 60–90 mins post injection (at least p<0.05). Overall F(3,17) = 7.316, p = 0.0023.

**Cold hypersensitivity.** In males (Fig 3C), 50 mg/kg induced significant decrease in acetone responses at 60 min (p = 0.0068) and 90 min post injection (p = 0.0448) compared to vehicle; 20mg/kg was effective at 60 min post injection (p = 0.0403). In females (Fig 3D), 50 mg/kg reduced acetone responses at 60 min post injection compared to vehicle (p = 0.0107).

**Dose response.** MPEs for each dose was calculated and used to determine A50 for each test (Fig 4). For tactile hypersensitivity, BCP A50s were 22.61 mg/kg in males (Fig 4A) and 35.22 mg/kg in females (Fig 4B). MPE at 60 mins for 50 mg/kg BCP (highest dose used) was 72.29% (males) and 61.32% (females). For cold hypersensitivity, BCP A50s were 19.03 mg/kg in males (Fig 4C) and 20.70 mg/kg in females (Fig 4D). MPE at 60 mins for 50 mg/kg BCP was 60.03% (males) and 68.67% (females).

## Experiment 2: Co-injection of CBD and BCP enhances reduction in tactile and cold hypersensitivity

Next, we evaluated the effect of co-administration of CBD and BCP using their respective A50 doses for each test to assess for possible synergistic effects. The dose ratios for CBD and BCP were calculated from the approximate individual CBD:BCP A50s. Dose combinations tested were the initial A50 doses of CBD and BCP, ½ the A50 doses, and ¼ the A50 doses.

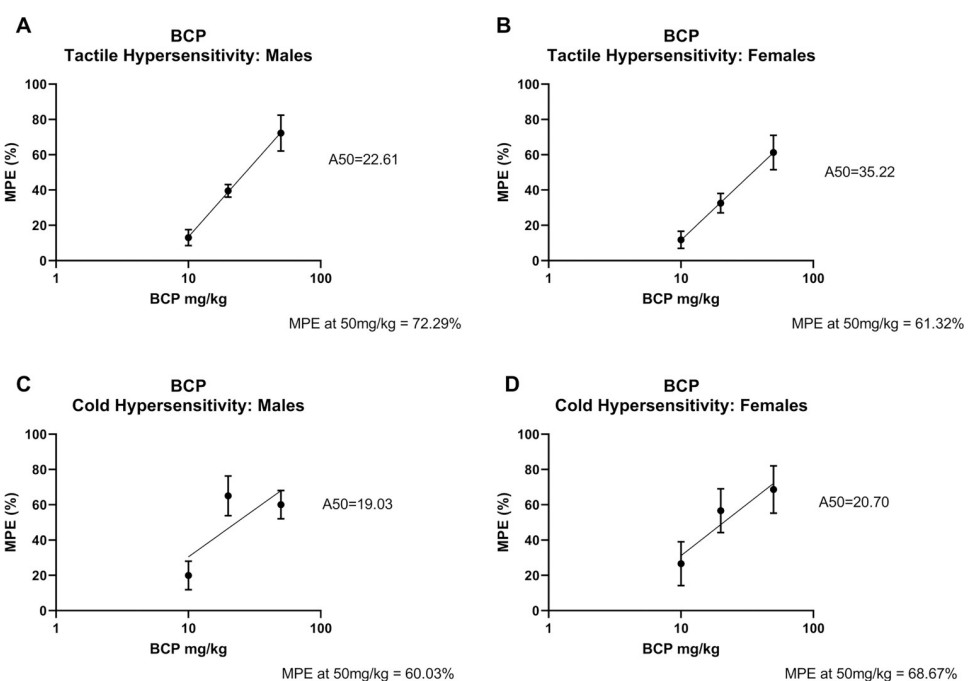

**Fig 4. Dose response curves for BCP antinociceptive effects in male and female rats.** Dose response curves for the effect of BCP on mechanical paw withdrawal threshold in A) males and B) females and acetone responses in C) males and D) females. A50 for each treatment group is displayed. MPE for the highest dose of BCP is indicated as well. Data are shown as % maximal possible effect (% MPE) ± SEM.

**Tactile hypersensitivity.** In males (Fig 5A), the highest concentration of CBD/BCP using the respective A50 doses in combination was 3 mg/kg CBD and 22 mg/kg BCP (dose ratio 3:22). The dose 3:22 was effective at 30–120 mins post injections, with maximum effects observed at 60 min and 90 min (all p<0.0001 compared with vehicle). A strong effect was also observed for the 1.5:11 dose, peaking at 60 min post injection (p<0.0001), with p = 0.0005 at 30 min and p = 0.0462 and p = 0.0164 at 90 and 120 mins post injection, respectively, compared to vehicle. The lowest dose combination of 0.8:5.5 was also effective compared to vehicle at 60 min (p<0.0001) and 90 min post injection (p = 0.0001). Overall F(3,16) = 34.10, p<0.0001. For context and comparison with clinically utilized agents, gabapentin, which is widely used as a first line treatment for neuropathic pain including SCI, results in dose-related attenuation of tactile hypersensitivity in this SCI model, with effects of the highest dose (100 mg/kg) comparable to the combined CBD/BCP 3:22 dose in male rats [41]. Morphine is also comparatively effective at 3 mg/kg, but develops rapid tolerance and has high misuse and side effects risks [43].

In females (Fig 5B), a similar trend was observed. Using the calculated A50s of the individual drugs in females, the highest starting concentration for the combination was 7 mg/kg CBD and 35 mg/kg BCP (7:35 dose ratio). This dose combination significantly reduced mechanical hypersensitivity at 30–120 mins post injection (at least p<0.01 compared with vehicle during that time). The tested lower combination doses were also effective at 30–120 mins (at least p<0.05 compared with vehicle). Overall F(3,16) = 11.58, p = 0.0003.

**Cold hypersensitivity.** In males (Fig 5C), the starting concentration was determined as 1 mg/kg CBD and 20 mg/kg BCP (1:20 dose ratio) based on Experiment 1 results. All tested combination doses, including the ¼ dose combination induced significant reduction in responses to acetone stimulation starting at 30 min post injection and lasting up to

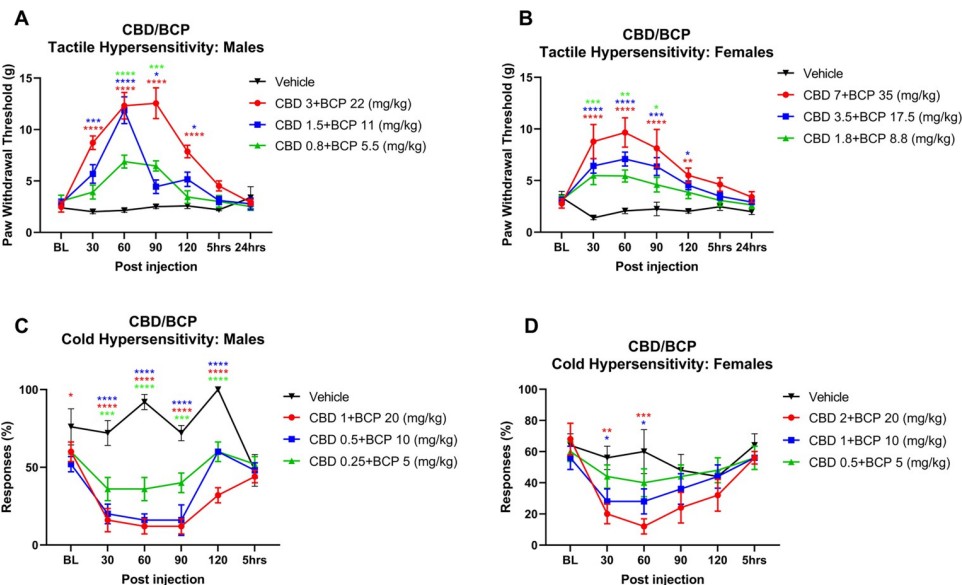

**Fig 5. Time course of CBD and BCP in combination on SCI pain responses in males and female rats.** Time plots of the effect of (A) CBD plus BCP on mechanical PWT in males; (B) CBD plus BCP on PWT in females; (C) CBD plus BCP on acetone responses in males; (D) CBD plus BCP on acetone responses in females (n = 5 per treatment group). All doses and dose ratios were determined from previous individual dose-response analysis as described in the text. *, **, ***, **** denote p < 0.05, 0.01, 0.001 and 0.0001 compared to vehicle for each treatment group.

120 mins (at least p<0.001 compared with vehicle). The most potent effects were observed for the 1:20 and 0.5:10 doses at 30–90 mins post injection with p<0.0001. Overall F(3,16) = 41.51, p<0.0001. In females (Fig 5D), the strongest effect on reducing cold hypersensitivity was observed for their highest combination doses (2 mg/kg CBD: 20 mg/kg BCP) at 30 min (p = 0.0067) and 60 min (p = 0.0002) post injection. The 1:10 dose was also effective at 30–60 mins (p = 0.0325 and 0.0126 respectively). Overall F(3,16) = 2.531, p = 0.0334.

**Dose response.** In order to determine the A50s for the drug combinations, the MPEs for each drug/test/sex were determined (Fig 6). A50 values for males were 1.06 mg/kg and 7.91 mg/kg for CBD and BCP respectively for tactile hypersensitivity (Fig 6A), and 0.26 mg/kg and 5.12 mg/kg for CBD and BCP respectively for cold hypersensitivity (Fig 6C), when used in combination at the selected dose ratios as determined from Experiment 1. In females, A50 values were 5.60 mg/kg and 28.32 mg/kg for CBD and BCP respectively for tactile hypersensitivity (Fig 6B), and 0.69 mg/kg and 6.90 mg/kg for CBD and BCP respectively for cold hypersensitivity (Fig 6D). MPEs for the highest CBD and BCP combination doses were over 80% in each test/sex, except for tactile hypersensitivity in females (MPE = 56.37%).

**Experiment 2: Synergistic effect of CBD/BCP for cold hypersensitivity.** The individual A50 values of CBD and BCP were used to plot the theoretical line of additivity of the combined drug administration for each of the sets (Fig 7). The experimental A50 values obtained from the combinations were plotted on these to determine additivity or potential synergism or antagonism. This analysis showed that, for tactile hypersensitivity, the effects of the CBD/BCP combinations were additive in both males (p = 0.083) (Fig 7A) and females (p = 0.236) (Fig 7B). For reducing cold hypersensitivity, the effects of the CBD and BCP co-administration were synergistic (males: p = 0.024, Fig 7C; females: p = 0.021, Fig 7D).

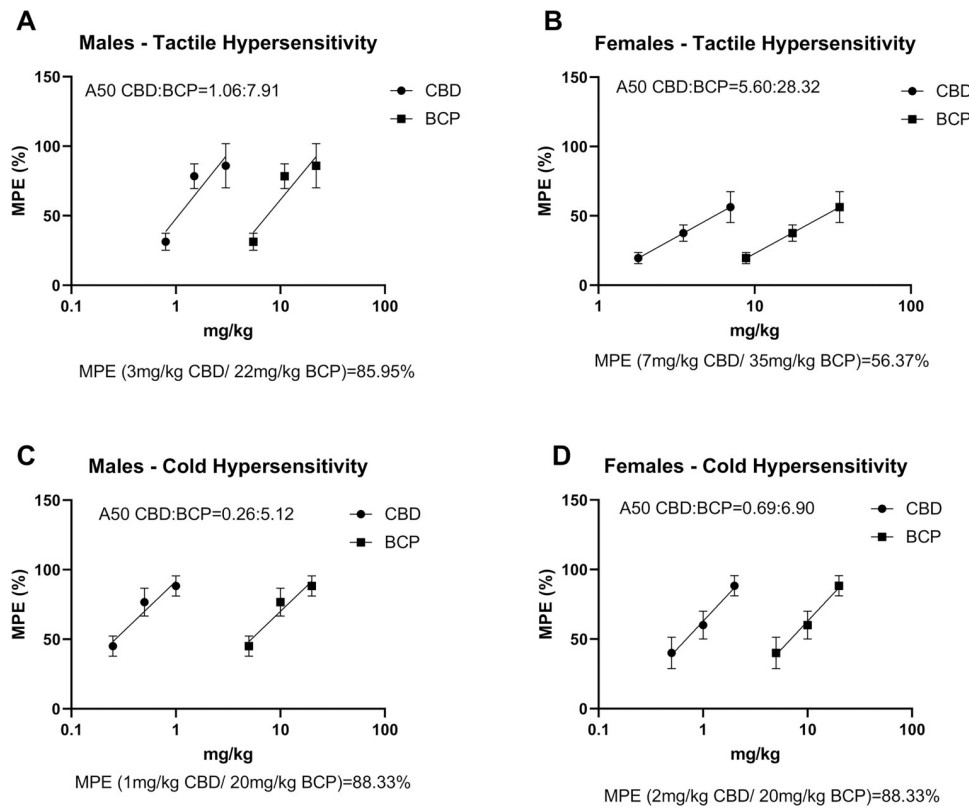

**Fig 6. Dose response curves for antinociceptive effects of CBD and BCP in combination in males and females.**
Dose response curves for the effect of CBD:BCP on mechanical paw withdrawal threshold in A) males and B) females and acetone responses in C) males and D) females. A50 for each drug in combination is displayed for each test. %MPE for the maximal doses of both drugs in combination is shown for each test. Data are shown as % maximal possible effect (% MPE) ± SEM.

## Experiment 3: Lack of significant side effects of CBD and BCP

To test for adverse effects of the antinociceptive CBD/BCP combination, we examined the traditional cannabinoid side effects profile. Intact rats were used for this, due to locomotor limitations of rats with SCI precluding full side effects analyses in rotarod and bar tests. A subset of SCI rats were evaluated using limited outcome tests. For all side effects testing, animals were administered the highest therapeutic doses 7:35 mg/kg CBD:BCP in order to detect any adverse effects of the combination. As a positive control, we also compared the side effects profiles of the combination to the mixed CB1/CB2 receptor agonist WIN 55212–2.

In both males and females, no significant differences were observed for rotarod latency or catalepsy in animals that received the CBD/BCP combination compared to baseline. In intact males, the CBD/BCP combination did not evoke any apparent detrimental effects in the rotarod test, with all latency values unchanged throughout the test. In contrast, WIN 55,212–2 injection led to decreased fall latency at 30–120 mins post injection compared to pre-injection baseline (p at least <0.01 throughout this time course). There were significant differences between treatments with overall $F(1,10) = 23.25$, $p = 0.0007$ (Fig 8A). In intact females, fall latencies in the rotarod test were overall comparable between treatment groups (Fig 8B). There was a transient non-significant drop in fall latency in the WIN 55212–2 group at 30 min post injection. Overall $F (1,10) = 3.60$, $p = 0.0870$. We observed a transient increase in body temperature at 2 hours post CBD/BCP injection ($p = 0.0151$ vs baseline) in intact males (Fig 8C);

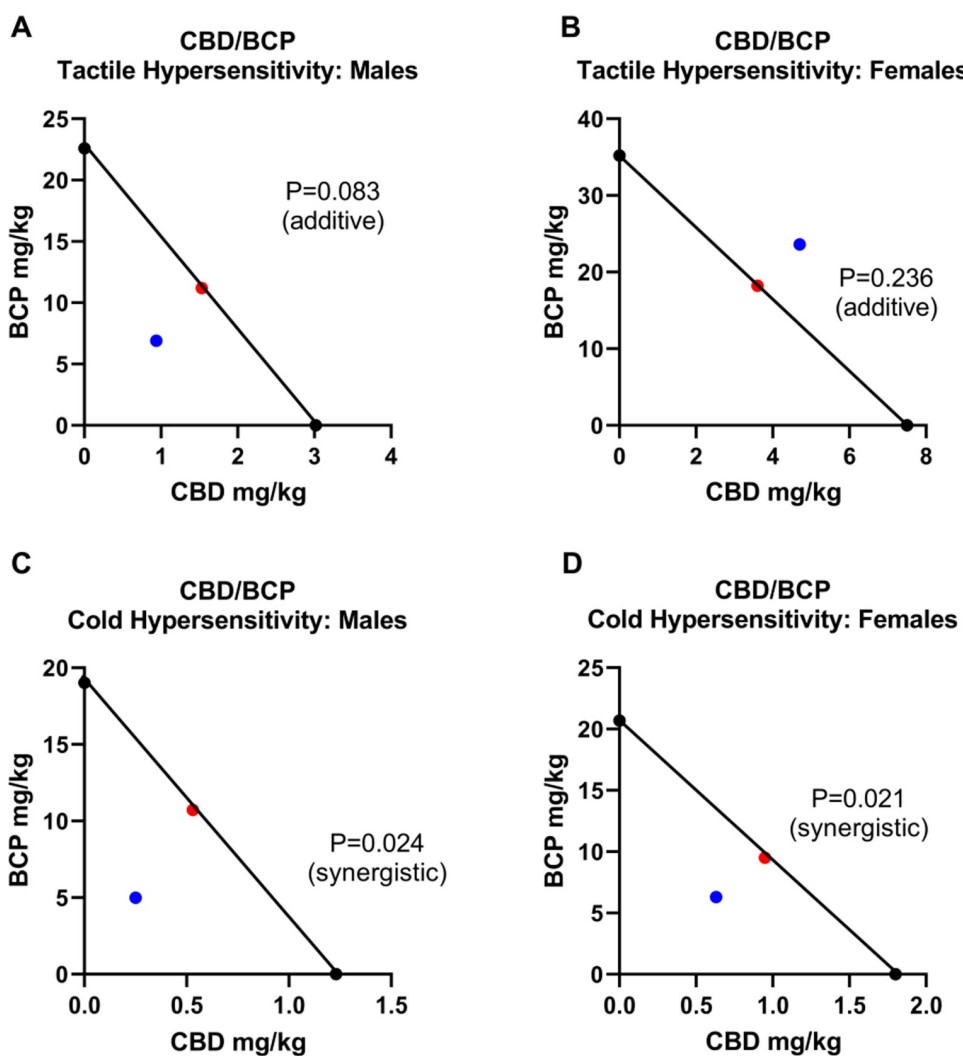

**Fig 7. Isobolographs of CBD and BCP antinociceptive effects in combination in males and female rats.**
Isobolographic analyses for combination CBD and BCP treatment on mechanical paw withdrawal threshold in A)
males and B) female rats and acetone responses in C) males and D) female rats. P values and the effect are indicated.

overall F (1,10) = 5.669, p = 0.0385. A similar trend was observed in intact females, but this
was not statistically significant (Fig 8D); overall F (1,10) = 0.3320. No effect of CBD/BCP was
observed in the catalepsy bar test in either male or female non-injured rats. In contrast, WIN
55,212–2 induced significant catalepsy starting 30 min post-injection through 5 hours in male
rats (p = 0.0099, 0.0049 and p<0.0001 respectively vs baseline), and strong differences between
groups were observed with overall F(1,9) = 63.23, p<0.0001 (Fig 8E). In females, WIN 55,212–
2 injection also caused significant catalepsy 30 min-5 hours post injection (with p value at least
<0.05 compared with pre-injection baselines, and significant differences between the drug
treatments (overall F(1,10) = 31.09, p = 0.0002; Fig 8F).

In SCI animals, no significant effects of the CBD/BCP combination were observed in body
temperature or bar tests (p = 0.8650 compared with pre-injection baseline; S1 Fig). However,
significant hypothermic effects of WIN 55212–2 were observed in this group from 30 min– 2
hrs post-injection in both males and females (p at least <0.01). There was also a transient (at
30 min) significant enhanced catalepsy behavior in male SCI rats (p = 0.0418).

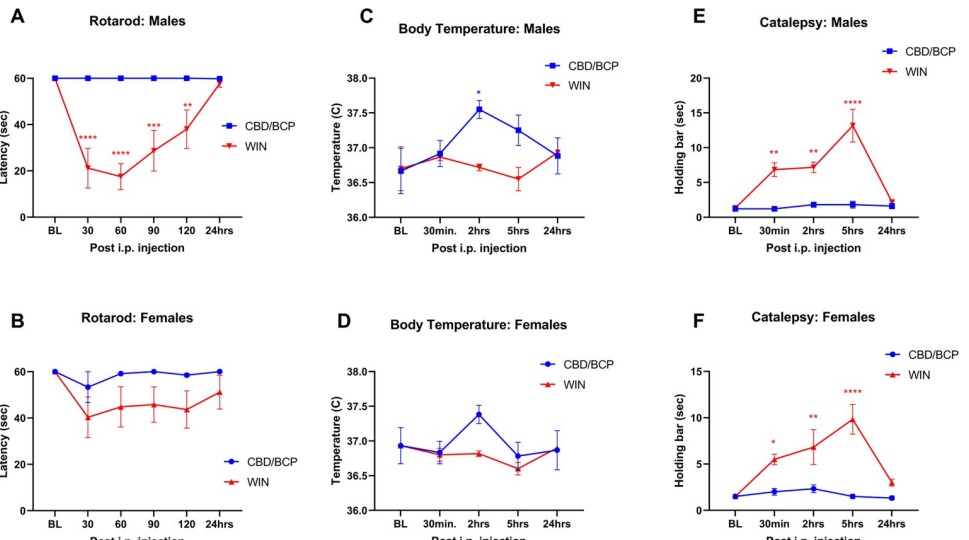

**Fig 8. Side effects profiles for CBD and BCP combination in male and female rats.** Time course showing the effect of maximum utilized antinociceptive dose combination of CBD (7 mg/kg) and BCP (35 mg/kg) compared with WIN 55212–2 (3 mg/kg) on rotarod latency in A) males and B) females, body temperature in C) males and D) females, respectively, and catalepsy bar latency in E) males and F) females (n = 6 per treatment group). Animals received a subcutaneous injection of WIN 55212–2 or an intraperitoneal injection and single oral administration of CBD:BCP following baseline determinations. *, **, ***, **** denote p < 0.05, 0.01, 0.001 and 0.0001 compared to baseline for each treatment group.

## Experiment 4: Attenuation of antinociceptive effects of the CBD/BCP combination by selective antagonists suggests interaction at the CB1 receptor

We examined the potential contributing role of CB1, CB2 or opioid receptor interactions on the antinociceptive actions of the combined CBD and BCP by pre-treating animals with selective receptor antagonists AM251, AM630, or naloxone 30 minutes prior to co-administering CBD/BCP at their determined A50 combination doses.

The CB1 antagonist AM251 strongly attenuated the antinociceptive effects of CBD/BCP on tactile hypersensitivity in males (Fig 9A) at 60–120 mins post injection (at least p<0.01; overall F(3,17) = 10.37, p = 0.0004). In females (Fig 9B), AM251 also strongly attenuated the antinociceptive effects of CBD/BCP at 60–120 mins post injection (at least p<0.05, overall F(3,16) = 23.46, p<0.0001). AM251 also strongly attenuated the antinociceptive effects of CBD/BCP on cold hypersensitivity in males (Fig 9C) at 30–120 mins post injection (at least p<0.05; overall F(3,17) = 9.879, p = 0.0005). This however was not observed for cold hypersensitivity in females (overall F(3,16) = 1.151, p = 0.3588). The other antagonists tested only showed transient and modest effects in some of the groups (e.g. AM630 at 60 min on tactile hypersensitivity in males; naloxone at 60 min on cold hypersensitivity in females), but were overall ineffective in reducing the robust antinociceptive effects of the CBD/BCP combination in SCI rats.

Since the apparent significant contribution of CB1 receptor-mediated effects of CBD/BCP combination was somewhat surprising, as neither of these agents have been thought to act via CB1, an additional comparison was done in retrospect with purified CBD in case of trace additional cannabinoids in the OTC CBD oil contributing to its antinociceptive effects. Results from this pilot comparison are shown in S2 Fig. Findings suggest potential modestly increased antinociceptive effects of CBD oil over pure CBD in males, but this was not significant likely due to the high degree of variability in the purified CBD groups.

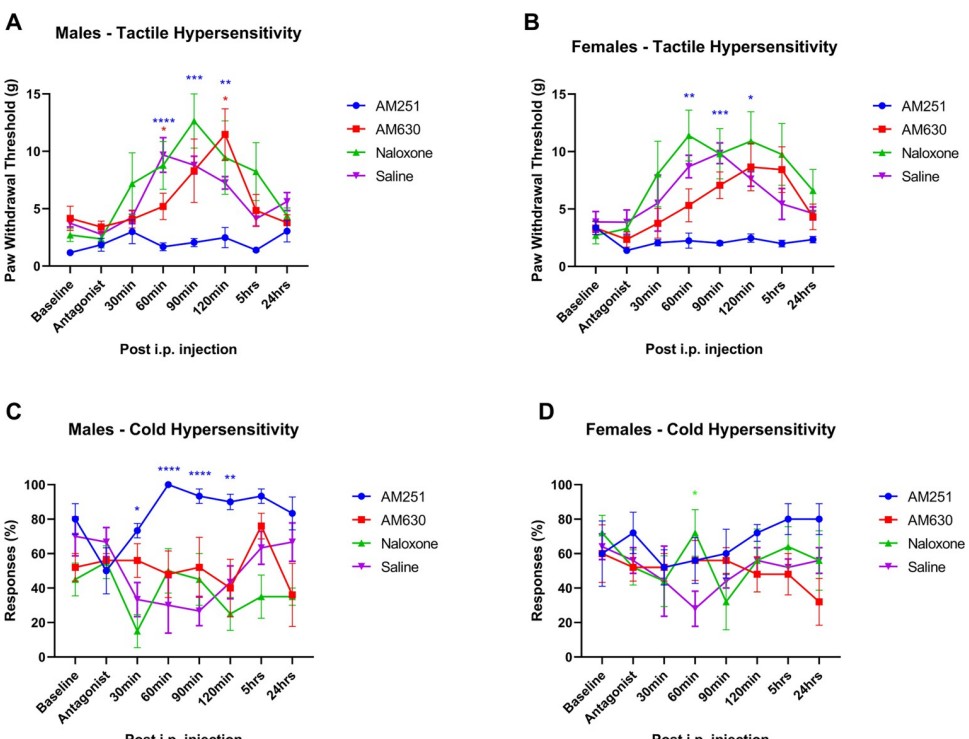

**Fig 9. Effect of selective antagonists on the antinociceptive effects of CBD and BCP combination.** Time plots of the antinociceptive effects of combined CBD (0.5, 1.5 mg/kg) and BCP (10, 15 mg/kg) following antagonist administration: AM251 (3 mg/kg), AM630 (1 mg/kg) or mu-opiod receptor antagonist naloxone (5 mg/kg) on mechanical PWT in A) males and B)females, and acetone responses in C) males and D) females, respectively (n = 5 per treatment group). Animals received a subcutaneous injection of the antagonist or saline 30 min prior to CBD/BCP administration in SCI rats. *, **, ***, **** denote p < 0.05, 0.01, 0.001 and 0.0001 (color coded), compared to vehicle for each treatment group.

## Experiment 5: Morphine seeking behavior is attenuated by CBD/BCP

To examine whether repeated CBD/BCP administration may reduce ongoing basal pain and consequently reduce opioid-seeking behavior, we used a low dose of morphine (2 mg/kg) as an analgesic reinforcing agent. 2x A50 doses were used for this part of the study to assure maximal antinociceptive benefits from the CBD/BCP in rat males SCI prior to morphine exposure. In non-injured intact animals, this low dose of morphine was not reinforcing and morphine CPP did not develop (Fig 10). In SCI animals not being treated with CBD/BCP, significant CPP to morphine developed, in comparison with SCI animals paired with saline only (p = 0.0078). In contrast, this was attenuated, with no significant morphine CPP observed in rats that had received CBD/BCP administration during their morphine conditioning. In addition, there were no significant differences between the CBD/BCP SCI group and the uninjured morphine group.

## Discussion

This study demonstrated that both CBD and BCP individually reduce hypersensitivity in males and females in a rat spinal cord injury pain model. Further, the co-administration of CBD:BCP synergistically attenuated cold hypersensitivity in both males and females with additive effects seen for reducing tactile hypersensitivity in males. Minimal cannabinoidergic-like

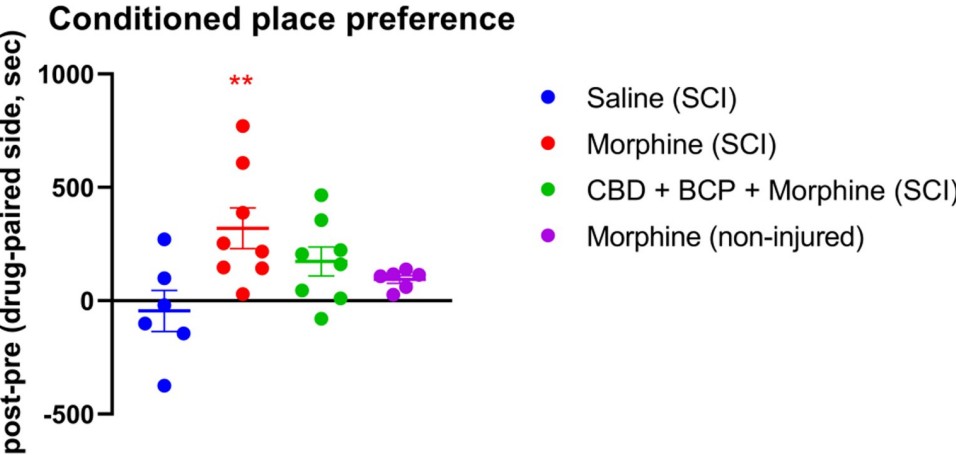

**Fig 10. Conditioned place preference measurements.** Data are shown as differences in time spent on the non-preferred side after conditioning minus pre-conditioning. Morphine (2 mg/ kg) was used as the conditioning analgesic agent in SCI or non-injured male rats; saline vehicle was used as conditioning control in SCI animals. Animals in the CBD/BCP group received CBD/BCP 1 hour prior to morphine during CPP training. ** denotes $p < 0.01$ compared to saline-paired SCI group.

side effects were observed by the combination. Together, findings from this study suggest that these non-psychoactive cannabis components may be an effective and readily attainable option for managing challenging neuropathic pain resulting from spinal cord injury.

Chronic pain following SCI is estimated to occur in up to 70% of patients, with at least one-third of patients rating it as so severe that it is their primary impediment to participation in daily activities and social well-being [1,60–63]. Although rigorously controlled studies have not been conducted for SCI pain, anecdotal reports from SCI patients with chronic pain have reported substantial pain relief from marijuana and whole plant medicinal extracts, suggesting the possibility that cannabinoids may be of particular value as a treatment option for this indication [64–68]. Cannabinoids have been shown to be effective in attenuation of pain-related behaviors in a wide variety of rodent inflammatory and peripheral neuropathy models, primarily via interaction with peripheral or spinal nociceptors [14,16,23,58,69–74]. However, there have thus far been limited studies evaluating the effects of naturally-derived cannabis components in preclinical SCI pain models [75,76] and only a few investigating synthetic CB1/2 agonists for this indication in preclinical models, primarily from work on our laboratory [41,43,77]. Further, strong CB1 agonists can additionally mediate undesirable CNS effects with systemic administration. Since the cannabis plant produces a wealth of cannabinoid compounds and terpenes acting via distinct mechanisms, beneficial analgesic effects in the absence of undesired side effects may be possible to achieve due to additive or synergistic contribution of several complementary components. Thus the goal of the current study was to explore the combination of two predominant and readily available OTC cannabis components with good safety and purportedly distinct mechanistic profiles to target SCI pain.

Results from this study showed that systemic administration of CBD or BCP individually can dose-dependently reduce SCI-related hypersensitivity in both male and female rats. Systemically administered CBD has shown dose-related moderate effectiveness in other rodent neuropathic pain models [69,70,78]. In the SCI model, CBD appeared more effective in reducing cold hypersensitivity than tactile hypersensitivity in both sexes, with more robust and prolonged attenuation of SCI-induced acetone responses, particularly at the higher doses (3–5 mg/kg). Effects on tactile hypersensitivity were more modest and short-lived, with only partial

attenuation at the highest dose. Systemic BCP alone also reduced SCI tactile and cold hyper-sensitivity, consistent with studies in other neuropathic pain models [12,20,26]. Following BCP administration we observed dose-dependent antinociceptive effects for both sexes. Although anti-allodynic effects were observed with BCP, this agent alone showed overall lower efficacy and shorter duration compared to CBD in reducing hypersensitivity in both sexes, even at the highest doses.

Upon coadministration of CBD and BCP, we observed dose-dependent reduction in SCI mechanical and cold hypersensitivity. Using isobolographic analysis, findings revealed that there was an additive effect in reducing mechanical hypersensitivity in males and a synergistic effect in reducing cold hypersensitivity across both sexes. Changes in MPE indicated that the coadministration of these two agents can improve the potency and efficacy of both CBD and BCP. In females, we observed an increase in the efficacy of CBD in reducing cold hypersensitivity (from %MPE = 73.33% individually to 88.33% in combination at highest doses), as well as an increase in CBD potency from A50 1.8 mg/kg to A50 0.69 mg/kg and increased BCP potency from A50 20.7 mg/kg to 6.9 mg/kg. In males, CBD maintained similar high efficacy in reducing cold hypersensitivity both when individually administered and in combination with BCP, but at substantially lower doses of both agents (from A50 1.23 mg/kg CBD and 19.03 mg/kg BCP with individual administration to A50 0.26 mg/kg CBD and 5.1 mg/kg BCP in combination). The combination also increased both efficacy and potency in reducing tactile hypersensitivity in males (CBD A50 from 3.02 mg/kg to 1.06 mg/kg; BCP A50 from 22.6 mg/kg to 7.9 mg/kg), and reaching nearly full reversal of SCI induced tactile hypersensitivity with the highest dose combination.

Interestingly, the effects of both compounds and the combination appeared less effective in females compared to males notably in reducing SCI induced tactile hypersensitivity. Numerous differences were observed throughout the study, including the duration of anti-allodynic effects of CBD individually and in combination with BCP, the potency and efficacy of CBD and BCP alone, and especially in the CBD/BCP combination. While SCI tactile hypersensitivity could be nearly completely reversed in males receiving the combination treatment, only partial attenuation was achieved in females despite the higher doses of both CBD and BCP used in the latter. These data suggest that there is some reduced effectiveness of cannabinoids in reducing SCI tactile hypersensitivity in females, both in response to CBD or BCP individually, in particular in response to the CBD/BCP combination, and underscore the importance of considering potential sex differences when developing cannabinoid-pain reducing strategies. Sex differences in responses to cannabinoids, including antinociceptive and locomotor effects in rodent models have been observed in numerous previous reports [27,79–83]. This has been hypothesized to result from differences in cannabinoid absorption, distribution, and metabolism, expression and contribution of CB1 and CB2 receptors and the endocannabinoid system, and/or interactions with gonadal hormones. However, findings have been inconsistent, with some reporting greater, lower, or equivalent antinociceptive effects in females vs males, depending on the etiology, outcome measure, and phenotype. Interestingly, in preclinical rodent acute and inflammatory pain models, females have been reported to show fairly consistently higher sensitivity than males to cannabinoid agonists such as THC and synthetic CB agonist CP55,940 [80,82,84], which seems in contrast with observations from the current study suggesting that the potential CB1-mediated antinociceptive effects are stronger in males than females. However, similar to our current findings, a CB1 agonist was found to require 30-fold higher dose in females than males in reducing mechanical hypersensitivity in a rat myositis model [81]. There have been fewer comparative studies with CBD alone, although a recent report showed no effect on females compared with males in a formalin test [85]. In addition, the stabilized CBD precursor cannabinoid, CBDA-ME (cannabidiolic acid methyl

ester) was shown to produce significant anti-hyperalgesic effects in males with peripheral neuropathic cuff injury, but had no effect on females [86]. With regard to BCP, more pronounced reduction in persistent inflammatory responses (formalin test) have been reported in males than females [27]. Thus, the current findings following these individual treatments are consistent with the limited preclinical literature. There have also been some reports using combination approaches, albeit not the current CBD/BCP combination. For example, using other *Cannabis sativa* terpenes in combination with synthetic CB agonist WIN55,212, boosted cannabinoid activity in acute pain responses (tail flick) was equivalently observed in all cases in males and females except for terpene linalool, which produced the same effects alone in both sexes, but greater potentiation of antinociceptive effects in males when in combination with the WIN55,212 [87]. There have been no comparative studies comparing sexually dimorphic effects of cannabinoids on SCI pain. However, spinal cord injury has been shown to produce dramatic increases in the spinal endocannabinoid system early after SCI, including increased levels of endocannabinoids and CB1 and CB2 receptors which may be involved in early neuroprotective effects, followed by later reduced expression [88–90].

Changes in CB receptors in higher brain processing regions have also been observed following SCI [91]. There are also reported increases in spinal endocannabinoid and CB1 receptors in peripheral neuropathic pain models in male rats [92,93]. In a recent evaluation of sex-related differences in a chemotherapy model of neuropathic pain, numerous differences were found in the DRG endocannabinoid components [94]. Similarly, in an orofacial myositis model, significant upregulation of CB1 receptor mRNA levels were found in trigeminal ganglia of male but not female rats, in parallel with a markedly reduced mechanical hypersensitivity by a selective CB1 agonist in males compared with females [81]. The latter was attributed to a testosterone role in the upregulation of CB1 receptors following myositis. Thus, the current observation of potential sex-related distinctions in responsiveness to cannabinoids may result from differences in endogenous cannabinoid system regulation following SCI. There are also emerging interesting sexually dimorphic T-cell differences in males but not females in parallel with improved neuropathic mechanical hypersensitivity following CBD and THC [95]. Further study of these potential differences will be critical moving forward towards clinical application.

To determine potential cannabinoidergic pharmacologic mechanisms mediating the effects of combined CBD and BCP, we pretreated animals with either CB1-selective antagonist AM251 or CB2-selective antagonist AM630 prior to administration of antinociceptive CBD/BCP combination. The opioid antagonist naloxone was also tested to assess for contribution of host opioid-mediated effects. Unexpectedly, the antinociceptive effects of CBD/BCP combination were nearly completely blocked by the CB1 antagonist (in both males and females for tactile hypersensitivity and males for cold hypersensitivity). Neither CB2 receptor antagonist nor opioid receptor antagonist resulted in substantial attenuation of effects. This finding suggests that, when used in combination, antinociceptive mechanisms involves CB1 receptor pathways. While our current study and previous literature showed that CBD and BCP can individually reduce hypersensitivity, their combined antinociceptive mechanism may differ from their purported individual mechanisms. Neither of the components of the CBD/BCP combination are thought to individually produce their antinociceptive effects via CB1 receptors according to the existing literature. In particular, BCP pharmacologic effects are nearly always attributed to CB2 receptor agonist activity, both in pain and other inflammatory tissue injury models [12,24–27,96–99]. There are additional potential neuropathic pain targets suggested for this terpene in addition to CB2 activation, but does not appear to involve CB1 when individually administered [100].

The current SCI model appears to involve both neuropathic and inflammatory components, as anti-inflammatory mediators in spinal cord tissue and surrounding CSF are markedly increased following this injury as reported previously in our lab [101,102]. The current SCI model appears to involve both neuropathic and inflammatory components, as anti-inflammatory mediators in spinal cord tissue and surrounding CSF are markedly increased following this injury as reported previously in our lab [101,102]. We have recently also found a reduction in a phantom limb pain model by CBD/BCP combined administration, along with reduced spinal inflammatory markers [103]. However, CB2 mechanisms are not likely to be the primary contributor to this chronic SCI pain, as previous findings in our lab have shown that selective CB1 antagonists, but not selective CB2 antagonists, block the analgesic effects of synthetic mixed CB1/CB2 agonists WIN 55,212–2 or CP 55,940 in this model [42,104]. In addition, anti-inflammatory agents have only modest beneficial effects in reducing clinical chronic SCI pain.

Thus, there is likely an additional contributing mechanism in the current observed robust effects of the combined CBD-BCP treatment. These results suggest the possibility that there are spinal cord injury-induced changes in cannabinoidergic pain processing leading to increased sensitivity to dorsal horn CB1 mediated effects. Previous findings in our lab have shown that SCI pain is particularly and uniquely sensitive to synthetic cannabinoid treatment in comparison with other pain models [42,43,104]. Thus, the current effective CBD/BCP combination may, directly or indirectly via downstream effects, activate novel upregulated antinociceptive CB1 sites or induce changes in endocannabinoid levels acting at CB1 receptors as described above. This interesting observation will be pursued in future studies. As a caveat, the original plan of the study did not include antagonist evaluations of the separate CBD and BCP components, as their likely mechanisms had already been reported in the literature and the CB1 dependence of the combined CBD/BCP had not been anticipated; thus, possible individual interactions with CB1 receptors cannot be ruled out. However, in light of this unexpected finding suggesting a CB1 receptor role in the combined CBD/BCP effect, we initiated pilot *in vitro* internalization assays to indicate whether this may involve a direct effect on CB1 receptors. Using CB1 receptor expressing cells, results suggested that CBD alone produces little effect, and BCP alone produces marginal CB1-activated receptor internalization, while the combination of CBD and BCP appears to enhance CB1 internalization, and this is blocked by CB1 antagonist AM251 (S3 Fig). Thus, it is possible that these drugs can allosterically facilitate one another's effect on CB1 receptors, and further supports a contribution of CB1 receptor activation, at least in part, to the observed SCI antinociceptive effects.

Another possibility that may account for the apparent key role of CB1 receptors is the presence of trace amounts of other cannabinoids found in OTC CBD oils that may act via CB1 receptors. The CBD Gold Oil used in the current study contains 0.04% CBD-V, 0.01% cannabigerol, 0.03% cannabinol, and 0.07% cannabichromene in addition to 5.45% CBD (undetectable delta-9 THC, delta-8 THC, THC-V, THC-A, cannabigerol-A, and CBD-A), according to the certificate of analysis (Koodegras; Millcreek, UT). While the aim of the study was to evaluate readily accessible CBD oil, a pilot comparison with purified CBD provided via the NIDA Drug Supply Program, showed essentially comparable albeit marginally higher antinociceptive effects of the same doses of CBD oil. Although unlikely to account for the robust observed CB1 effects of the CBD/BCP combination, this possibility will be pursued further in future studies.

In conclusion, the current findings indicate that the combination of readily accessible non-psychoactive cannabis components CBD oil and BCP may be particularly effective in reducing neuropathic pain resulting from spinal cord injury. In addition, cannabinoid-like side effects were minimal using this combination. Further, the observed decrease in opioid-seeking behavior suggest that this treatment may be useful as a supplemental therapeutic to reduce opioid

needed for effective pain management. Together, these findings are supportive of the beneficial effects of combining cannabis components in the armamentarium for chronic pain management.

## Supporting information

**S1 Fig. Side effects profiles for CBD and BCP combination in male and female rats with SCI.** Time course curves showing the effect of maximum utilized antinociceptive dose combination of CBD (7 mg/kg) and BCP (35 mg/ kg) compared with WIN 55212–2 (3 mg/kg) on (A) body temperature and (B) catalepsy bar latency in males and females respectively (n = 6 per treatment group). Animals received a subcutaneous injection of WIN 55212–2 or an intraperitoneal injection and single oral administration of CBD:BCP following baseline measurements at 4 weeks post SCI surgery. \*, \*\*, \*\*\*, \*\*\*\* denote p < 0.05, 0.01, 0.001 and 0.0001 compared to baseline for each treatment group.
(TIF)

**S2 Fig. Dose response comparison between antinociceptive effects of purified CBD with CBD oil (OTC CBD) antinociceptive in male and female rats.** Dose response curves for the effect of CBD formulations on mechanical PWT in A) males and B) females and acetone responses in C) males and D) females. Data are shown as % maximal possible effect (% MPE) ± SEM.
(TIF)

**S3 Fig. Internalization of CB1 receptor by vehicle, CB agonist WIN 55,212–2 (3μm), CBD (0.5mg/ml), BCP (4mg/ml), CBD/BCP (0.5mg/ml:4mg/ml), AM251 (1mg/ml).** CB1 receptor is depicted in red, located in the cytoplasmic membrane when not activated, forming a rim around the cell. Upon activation receptor is translocated into cytoplasm in the form of small clusters with different density. Nuclei labeled with DAPI are blue.
(TIF)

## Acknowledgments

The authors thank Ms. Shirah Aguayo and Ms. Yousra Erritouni for excellent technical assistance.

## Author Contributions

**Conceptualization:** Stanislava Jergova, Jacqueline Sagen.

**Data curation:** Stanislava Jergova.

**Formal analysis:** Anjalika Eeswara.

**Investigation:** Anjalika Eeswara, Amanda Pacheco-Spiewak.

**Methodology:** Anjalika Eeswara, Amanda Pacheco-Spiewak, Stanislava Jergova.

**Project administration:** Stanislava Jergova.

**Supervision:** Stanislava Jergova, Jacqueline Sagen.

**Validation:** Amanda Pacheco-Spiewak.

**Writing – original draft:** Anjalika Eeswara, Amanda Pacheco-Spiewak.

**Writing – review & editing:** Stanislava Jergova, Jacqueline Sagen.

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
