## [Decision Letter · Decision Letter 0]

1 Nov 2022

PONE-D-22-27988Combined non-psychoactive Cannabis components cannabidiol and β-caryophyllene reduce chronic pain via CB1 interaction in a rat spinal cord injury modelPLOS ONE

Dear Dr. Jergova,

Thank you for submitting your manuscript to PLOS ONE. After careful consideration, we feel that it has merit but does not fully meet PLOS ONE’s publication criteria as it currently stands. Therefore, we invite you to submit a revised version of the manuscript that addresses the points raised during the review process.

Your manuscript is reviewed by two experts in the subject and both provided positive feedback with major comments. Please address those comments as appropriate. I would like to add a detailed discussion about the gender differences and possible hypothesis. 

We look forward to receiving your revised manuscript.

Kind regards,

Partha Mukhopadhyay, Ph.D.

Section Editor

PLOS ONE

Journal Requirements:

2. Please ensure that you refer to Figure 7 in your text as, if accepted, production will need this reference to link the reader to the figure.

Reviewers' comments:

Reviewer's Responses to Questions

**Comments to the Author**

1. Is the manuscript technically sound, and do the data support the conclusions?

Reviewer #1: Yes

Reviewer #2: Yes

2. Has the statistical analysis been performed appropriately and rigorously? 

Reviewer #1: Yes

Reviewer #2: Yes

3. Have the authors made all data underlying the findings in their manuscript fully available?

Reviewer #1: Yes

Reviewer #2: Yes

4. Is the manuscript presented in an intelligible fashion and written in standard English?

Reviewer #1: Yes

Reviewer #2: Yes

5. Review Comments to the Author

Reviewer #1: In the present manuscript Eeswaara et. al reported a study on the analgesic potential of cannabidiol (CBD) and β-caryophyllene (BCP) administered individually or in combination to rats subjected to spinal cord injury (SCI), whereas, chronic pain was induced by clip compression. The authors revealed that both phytocannabinoids induced a dose dependent reduction in tactile and cold hypersensitivity when administered separately to male and female rats with SCI. CBD and BCP co-administration displayed an enhanced dose dependent reduction in allodynic responses with synergistic effects observed for cold hypersensitivity in both sexes and additive effects for tactile hypersensitivity in males. Interestingly, the antinociceptive effects of the combined CBD-BCP treatment were not inhibited by either CB2 (AM630) or μ-opioid receptor antagonist (naloxone) pretreatment but, were almost completely blocked by the CB1 antagonist AM251.

The goals are clear, the manuscript is well-written and readable, however, there are some typos that should be corrected in the final version.

Nevertheless, there are major comments need to be addressed:

1. Antinociceptive effects of both BCP and BCP treatment as well as their combined administration represented less robust in females than males. What can be the possible explanation of this phenomenon? Is there any sex difference in the expression CB1 receptors in the dorsal segment of these rat’s lumbar spinal cord following SCI?

2. BCP is thought to be a CB2 receptor agonist and had been shown to play an anti-inflammatory role in several tissue injury models via activating CB2 receptor signaling (please see PMIDs: 22326488; 27379721; 28107775). Interestingly, the authors found that administration of AM630 (a CB2 antagonist) barely affected the effects of combined CBD-BCP treatment (Fig.9). However, the current SCI model is not entirely associated with chronic tissue inflammation. Did the authors study the anti-inflammatory effect of the combined CBD-BCP treatment in another neuropathic pain model (e.g.: carrageenan-induced inflammatory pain model) that involves inflammation and activation of inflammatory leukocytes? Please provide some data using another inflammatory pain model.

3. Did the authors investigate the CB1-dependence of CBD and BCP drugs when given separately? In the combined treatment setting, can the drugs facilitate allosterically one another’s effect on CB1 receptors?

Reviewer #2: Please find my comments below:

1. The manuscript is written well, the methods are adequately described, and the results are properly explained in the results and discussion.

2. Abstract: 'Minimal cannabinoidergic adverse side effects were observed with high doses of the combination.' Write either adverse effects or side effects.

3. 'A50' has not been defined in the manuscript as well as in the main text. It would be difficult for someone who is not familiar with this term.

4. The section 'Drugs administration' should be ahead of 'Experimental design'.

5. Results: Rewrite the title of each result in order to highlight the main message of that particular results.

6. How much the efficacy or analgesic potential of CBD and BCP alone and/or in combination can be compared with the approved analgesics (central and/or peripheral)? The authors need to report few experiments in the same model in order to answer this question?

6. PLOS authors have the option to publish the peer review history of their article (what does this mean?). If published, this will include your full peer review and any attached files.

Reviewer #1: No

Reviewer #2: **Yes: **Abhishek Basu

---

## [Author Response · Author response to Decision Letter 0]

21 Dec 2022

Dear Dr. Mukhopadhyay,

We would like to thank you and the reviewers for the positive response and helpful suggestions for improving our manuscript. We have addressed the reviewer suggestions in the revised manuscript (marked in red) as follows:

Editor comment: I would like to add a detailed discussion about the gender differences and possible hypothesis.

Response: We have revised this section of the discussion to include more in depth consideration of our observed gender differences and possible underlying mechanisms (p22-24, highlighted in red). We have also done an extensive literature search in the field for insights and comparisons with our findings, but there are thus far little in published reports regarding gender differences in spinal cord injury (SCI) pain responses to cannabinoids. However, our data does suggest that there is some reduced effectiveness of cannabinoids in reducing SCI tactile hypersensitivity in females, both in response to CBD or BCP individually, and in particular in response to the CBD/BCP combination. The reductions in cold allodynia were essentially comparable in males and females. These differences have been clarified in the revision. Since the most robust distinction observed between males and females was in response to the combination CBD/BCP, both in potency and efficacy in reducing SCI tactile hypersensitivity, and our findings suggest that the antinociceptive effects of this combination are mediated via CB1 receptors, a hypothesis is that there are sex differences in the CB1/endocannabinoid system in response to injury, as suggested by Reviewer 1 (below). Although there are no published reports showing this per se, we have incorporated findings that may support this hypothesis, and certainly should be pursued in future studies. 

Reviewer 1: Thank you for your supportive comments and interest, and insightful suggestions. We have addressed the major comments in the revised manuscript as follows: 

1. Antinociceptive effects of both BCP and BCP treatment as well as their combined administration represented less robust in females than males. What can be the possible explanation of this phenomenon? Is there any sex difference in the expression CB1 receptors in the dorsal segment of these rat’s lumbar spinal cord following SCI?

Response: We have done an extensive search to determine whether there are reported sex differences in the expression of CB1 receptors in the spinal cord following SCI. Although this has not been reported, there are reported changes in both CB receptors and other components of the endocannabinoid system following SCI. There have been some sex differences reported in other models of chronic pain (neuropathic and inflammatory) in the endocannabinoid system that may provide additional insights. These points as well as a more comprehensive comparison between male and female responses to cannabinoids have now been included in a detailed section in the revised discussion (p22-24). The underlying mechanisms of these interesting differences, while beyond the scope of the current study, warrant further investigation going forward.

2. BCP is thought to be a CB2 receptor agonist and had been shown to play an anti-inflammatory role in several tissue injury models via activating CB2 receptor signaling (please see PMIDs: 22326488; 27379721; 28107775). Interestingly, the authors found that administration of AM630 (a CB2 antagonist) barely affected the effects of combined CBD-BCP treatment (Fig.9). However, the current SCI model is not entirely associated with chronic tissue inflammation. Did the authors study the anti-inflammatory effect of the combined CBD-BCP treatment in another neuropathic pain model (e.g.: carrageenan-induced inflammatory pain model) that involves inflammation and activation of inflammatory leukocytes? Please provide some data using another inflammatory pain model. 

Response: We have emphasized this point and included the additional references above clearly indicating the BCP is generally considered as a CB2 receptor agonist in mediating anti-inflammatory and anti-nociceptive effects. The current SCI model appears to involve both neuropathic and inflammatory components, as anti-inflammatory mediators in spinal cord tissue and surrounding CSF are markedly increased following this injury as reported previously in our lab (Dugan et al., 2020, 2021). However, this is not likely to be the primary contributor to this SCI pain, as previous findings in our lab have shown that selective CB1 antagonists, but not selective CB2 antagonists, block the analgesic effects of synthetic mixed CB1/CB2 agonists WIN 55,212-2or CP 55,940 in this model (Hama and Sagen, 2007; Hama et al., 2014). Thus, there is likely an additional contributing mechanism in the current observed robust effects of the combined CBD-BCP treatment. In addition, anti-inflammatory agents have only modest beneficial effects in reducing clinical chronic SCI pain. With regard to data in other neuropathic pain models with inflammatory components, while outside the scope of the current study, we have recently found a reduction in a phantom limb pain model by CBD/BCP combined administration, along with reduced spinal inflammatory markers – however this work is in progress and has been presented in abstract form (Eeswara et al., 2022). These points have been added to the revised discussion (p25-26). 

3. Did the authors investigate the CB1-dependence of CBD and BCP drugs when given separately? In the combined treatment setting, can the drugs facilitate allosterically one another’s effect on CB1 receptors? 

Response: Since the original goal of the study was to determine whether the combined administration of two reported mechanistically distinct readily available Cannabis components could provide increased attenuation of SCI-related neuropathic pain, we had relied on the literature reports for individual mechanisms. Thus, CB1-dependence of CBD and BCP effects were not originally anticipated. This caveat in retrospect is now included in the revised manuscript. However, we were also intrigued after obtaining this unexpected result indicating CB1 involvement in the combined CBD-BCP effect, and thus had initiated pilot in vitro internalization assays to indicate whether this may involve a direct effect on CB1 receptors. Using CB1 receptor expressing cells, results suggested that CBD alone produces little effect, and BCP alone produces marginal CB1-activated receptor internalization, while the combination of CBD and BCP appears to enhances CB1 internalization, and this is blocked by CB1 antagonist AM251. Thus, it is possible that the drugs can allosterically facilitate one another’s effect on CB1 receptors, as suggested. It must be emphasized that these are preliminary data at this stage and needs to be further explored in future studies. We have added examples of this pilot study as an additional supplementary figure (p26-37, Supplemental Figure 3). 

Reviewer 2: Thank you for your positive comments suggested clarifications to enhance the readability of our manuscript. We have addressed the comments in the revised manuscript as follows: 

2. Abstract: 'Minimal cannabinoidergic adverse side effects were observed with high doses of the combination.' Write either adverse effects or side effects.

Response: The sentence has been updated according to the suggestion (p2, L37)

3. 'A50' has not been defined in the manuscript as well as in the main text. It would be difficult for someone who is not familiar with this term.

Response: We have added an explanation for “A50” into Experimental Design (p7, L146-147).

4. The section 'Drugs administration' should be ahead of 'Experimental design'.

Response: The section has been moved as suggested (p6)

5. Results: Rewrite the title of each result in order to highlight the main message of that particular results

Response: The titles have been re-written (highlighted in red font).

6. How much the efficacy or analgesic potential of CBD and BCP alone and/or in combination can be compared with the approved analgesics (central and/or peripheral)? The authors need to report few experiments in the same model in order to answer this question?

Response: We have utilized this model in a number of analgesics testing over the past several years, and clinically approved agents for SCI pain were originally tested in the pharmacologic validation of the model (Hama and Sagen, 2007). In particular, gabapentin, which is widely used as a first line treatment for neuropathic pain including SCI, results in dose-related attenuation of tactile hypersensitivity in this model, with effects of the highest dose (100 mg/kg) comparable to the combined CBD/BCP 3:22 dose (Hama and Sagen, 2007). Morphine is also comparatively effective at 3 mg/kg, but develops rapid tolerance and has high misuse and side effects risks (Hama and Sagen, 2009). These comparisons have now been added to revised results in the appropriate section (p15). 

Thank you again for the positive reviews of our manuscript. We feel that the suggested reviewer revisions have improved our manuscript and are looking forward to its publication in PLOS ONE.

---

## [Decision Letter · Decision Letter 1]

10 Jan 2023

PONE-D-22-27988R1Combined non-psychoactive Cannabis components cannabidiol and β-caryophyllene reduce chronic pain via CB1 interaction in a rat spinal cord injury modelPLOS ONE

Dear Dr. Jergova,

Thank you for submitting your manuscript to PLOS ONE. After careful consideration, we feel that it has merit but does not fully meet PLOS ONE’s publication criteria as it currently stands. Therefore, we invite you to submit a revised version of the manuscript that addresses the points raised during the review process.

Your manuscript was reviewed by the same experts and one of them raised a minor point, which require your attention. A quick editorial decision will be taken after satisfactory revision of the manuscript without sending to the reviewer.

We look forward to receiving your revised manuscript.

Kind regards,

Partha Mukhopadhyay, Ph.D.

Section Editor

PLOS ONE

Journal Requirements:

Reviewers' comments:

Reviewer's Responses to Questions

**Comments to the Author**

1. If the authors have adequately addressed your comments raised in a previous round of review and you feel that this manuscript is now acceptable for publication, you may indicate that here to bypass the “Comments to the Author” section, enter your conflict of interest statement in the “Confidential to Editor” section, and submit your "Accept" recommendation.

Reviewer #1: All comments have been addressed

Reviewer #2: All comments have been addressed

2. Is the manuscript technically sound, and do the data support the conclusions?

Reviewer #1: Yes

Reviewer #2: Yes

3. Has the statistical analysis been performed appropriately and rigorously? 

Reviewer #1: Yes

Reviewer #2: Yes

4. Have the authors made all data underlying the findings in their manuscript fully available?

Reviewer #1: Yes

Reviewer #2: Yes

5. Is the manuscript presented in an intelligible fashion and written in standard English?

Reviewer #1: Yes

Reviewer #2: Yes

6. Review Comments to the Author

Reviewer #1: The authors have improved the present manuscript considerably by incorporating suggestions and remarks by the reviewers. The figures have been adequately revised in order to meet with the general concept of the study. There is only one major comment though that need to be addressed:

1. A detailed descriptions is needed for the cell culture experiments in the materials and methods section explaining the cell type as well as the duration of treatment used for studying CB1 receptor internalization following combined CBD/BCP administration as well as the imaging technique. Based on the corresponding figure legend, the authors used a CB1 receptor expressing cell line, however, receptor internalization was revealed in a non-specific manner (labelled hemagglutinin). Due to the lack of reliable antibodies, the authors should use other approach (e.g.: GFP-tagged CB1 receptor expressing cell line) to provide mechanistic explanation on the synergistic effects of CBD and BCP on CB1 receptor internalization.

Reviewer #2: The authors addressed all the points raised by the reviewers. They have also incorporated required changes in the manuscript.

7. PLOS authors have the option to publish the peer review history of their article (what does this mean?). If published, this will include your full peer review and any attached files.

Reviewer #1: No

Reviewer #2: **Yes: **Abhishek Basu

---

## [Author Response · Author response to Decision Letter 1]

13 Feb 2023

Dear Dr. Mukhopadhyay,

We would like to thank you and the reviewers again for positive comments and suggestions. We have addressed the reviewer suggestions in the revised manuscript as follows:

Reviewer #1: The authors have improved the present manuscript considerably by incorporating suggestions and remarks by the reviewers. The figures have been adequately revised in order to meet with the general concept of the study. There is only one major comment though that need to be addressed:

1. A detailed descriptions is needed for the cell culture experiments in the materials and methods section explaining the cell type as well as the duration of treatment used for studying CB1 receptor internalization following combined CBD/BCP administration as well as the imaging technique. Based on the corresponding figure legend, the authors used a CB1 receptor expressing cell line, however, receptor internalization was revealed in a non-specific manner (labelled hemagglutinin). Due to the lack of reliable antibodies, the authors should use other approach (e.g.: GFP-tagged CB1 receptor expressing cell line) to provide mechanistic explanation on the synergistic effects of CBD and BCP on CB1 receptor internalization.

Thank you for the suggestion, we have added CB1 cell culture and internalization assay methods into Method section (lines 237-261).

The HEK293 cell line used in this section express CB1 receptor with an N-terminus hemagglutinin (HA) tag; therefore the HA antibody is specific for labeling the tagged CB1 receptor and tracking the internalization of the receptor. This approach was developed for this purpose as reported by other studies (e.g. Daigle et al., 2008; ref 54; Grimsey et al., 2008; ref 56) and has been used in previous work in our lab (Jergova et al., 2021; ref 57) as well. Although the current study included CB1 internalization as a supplemental pilot assessment, future work will be undertaken to further understand the cellular mechanisms underlying CBD and BCP synergism. 

Thanks to all reviewers for the positive reviews of our manuscript.

Responses to Editor:

We have uploaded raw data from behavioral and statistical evaluations into Dryad database, together with TIFF files for figures used as Supplementary file 3.

We have added a reference to Fig. 7 into the manuscript.

We had removed funding information from the manuscript.

---

## [Editor Report · Decision Letter 2]

27 Feb 2023

Combined non-psychoactive Cannabis components cannabidiol and β-caryophyllene reduce chronic pain via CB1 interaction in a rat spinal cord injury model

PONE-D-22-27988R2

Dear Dr. Jergova,

We’re pleased to inform you that your manuscript has been judged scientifically suitable for publication and will be formally accepted for publication once it meets all outstanding technical requirements.

Kind regards,

Partha Mukhopadhyay, Ph.D.

Section Editor

PLOS ONE
---

## [Editor Report · Acceptance letter]

3 Mar 2023

PONE-D-22-27988R2 

Combined non-psychoactive Cannabis components cannabidiol and β-caryophyllene reduce chronic pain via CB1 interaction in a rat spinal cord injury model 

Dear Dr. Jergova:

I'm pleased to inform you that your manuscript has been deemed suitable for publication in PLOS ONE. Congratulations! Your manuscript is now with our production department. 

Kind regards, 

on behalf of

Dr. Partha Mukhopadhyay 

Section Editor

PLOS ONE